# Body Composition and “Catch-Up” Fat Growth in Healthy Small for Gestational Age Preterm Infants and Neurodevelopmental Outcomes

**DOI:** 10.3390/nu14153051

**Published:** 2022-07-25

**Authors:** Laura E. Lach, Katherine E. Chetta, Amy L. Ruddy-Humphries, Myla D. Ebeling, Mathew J. Gregoski, Lakshmi D. Katikaneni

**Affiliations:** 1Division of Neonatology, Department of Pediatrics, Medical University of South Carolina, Charleston, SC 29425, USA; chetta@musc.edu (K.E.C.); ruddya@musc.edu (A.L.R.-H.); ebelingm@musc.edu (M.D.E.); katikalm@musc.edu (L.D.K.); 2Department of Public Health Sciences, Medical University of South Carolina, Charleston, SC 29425, USA; gregoski@musc.edu

**Keywords:** body composition, neonatal intensive care, neonatal nutrition, neurodevelopmental outcome, nutrition/growth, very low birth weight

## Abstract

To examine the growth and body composition of small for gestational age (SGA) and appropriate for gestational age (AGA) very low birth weight infants (VLBW) and their outpatient neurodevelopmental outcomes. From 2006–2012, VLBW infants (*n* = 57 of 92) admitted to the Neonatal Intensive Care Unit (NICU) had serial air displacement plethysmography (ADP) scans and were followed as outpatients. Serial developmental testing (CAT/CLAMS, Peabody Gross Motor Scales) and anthropometrics were obtained from *n* = 37 infants (29 AGA and 8 SGA) and analyzed via repeated measures analyses of variances. The percentage of body fat, percentage of lean mass, and weight gain were statistically significant between SGA and AGA groups at the first ADP assessment. There was no difference between the two groups in outpatient neurodevelopmental testing. Weight gain as “catch-up” body fat accrual occurs by 67 weeks of PMA. This catch-up growth is associated with normal SGA preterm neurodevelopment as compared to AGA preterm infants.

## 1. Introduction

Optimizing nutrition for preterm infants continues to be a challenge to curb postnatal growth faltering in the neonatal intensive care unit (NICU), especially in the small for gestational age (SGA) and very low birth weight (VLBW) population. Studies show growth of VLBW infants is associated with better neurodevelopmental outcomes [1,2,3,4]. Previous work on the body composition of the preterm infant associated with hospital weight gain [2] and fat-free mass gain while in the NICU with better neurodevelopmental outcomes across multiple studies [2,3,4,5,6]. This is of crucial significance as SGA infants are at risk for poor neurodevelopmental outcomes [7].

Prematurity has been linked to later metabolic disease given increased early adiposity [8,9]. Few studies have suggested that rapid growth may lead to worse metabolic outcomes in adulthood [10,11,12]. There may be short-term benefits to increased rapid weight gain in the domains of survival and neurodevelopment, but this is controversial [10]. 

The link between premature infant growth and neurodevelopment is observed in multiple studies [2,3,4,5]. Ehrenkranz et al. demonstrated that faster weight gain and head growth in the NICU was associated with higher cognitive and motor scores and reduced cerebral palsy at 18–22 months [2]. In-hospital gain of fat-free mass is associated with better neurologic and motor outcomes at one-year corrected age in VLBW infants as demonstrated by Ramel et al. [6]. Small for gestational age (<10th percentile) infants are at especially increased risk for delayed neurodevelopment [7]. Studies have linked improved neurodevelopmental outcomes of preterm infants with respect to post-discharge nutrition, however, do not correlate with the post-discharge body composition [13]. 

The Capute Scales Cognitive Adaptive Test/Clinical Linguistic and Auditory Milestone Scale (CAT/CLAMS) is a standardized tool to assess pediatric development. This test has been established with acceptable sensitivity and specificity to detect a neurodevelopmental delay in the premature population and correlates well with the Bayley Scales of Infant Development-II [14,15]. The CAT/CLAMS Developmental Quotients (DQ) and Peabody Developmental Motor scales are both utilized at the Medical University of South Carolina’s (MUSC) outpatient NICU follow-up clinic. As more premature infants are surviving into adulthood with advances in medical and nutritional therapy, long-term observational studies are needed to establish outcomes of neurodevelopment and optimal growth. Our study aimed to examine the growth and body composition of VLBW SGA and VLBW appropriate for gestational age (AGA) infants over time and their neurodevelopmental outcomes at a level IV NICU.

## 2. Materials and Methods

From 2006–2012, 366 infants admitted to MUSC’s level IV NICU underwent an air displacement plethysmography (ADP) using the PeaPod (Cosmed, Concord, CA, USA) as part of clinical care. After discharge, all VLBW infants were followed, outpatient. To clarify the association between neurodevelopmental outcomes and body composition, we chose to exclude several severe co-morbidities and sample a population of healthy preterm infants. A total of 92 VLBW SGA and VLBW AGA infants were selected for analysis after excluding necrotizing enterocolitis, retinopathy of prematurity stage 3, and grade III-IV intraventricular hemorrhage. A total of 57 infants had serial ADP assessments, while admitted to the MUSC NICU and as an outpatient. The average time of the first ADP assessment was 44 ± 8 weeks postmenstrual age (PMA) and the second ADP at 67 ± 6 weeks PMA. Infants receiving an ADP evaluation were not on mechanical or non-invasive ventilation and did not have intravenous fluids.

After discharge, infants were then followed serially at outpatient at MUSC’s high risk NICU clinic where neurodevelopmental testing (CAT/CLAMS and Peabody Gross Motor) was performed along with anthropometric measurements including height, weight and head circumference at each outpatient visit. Chronological and adjusted age CAT, CLAMS, and Gross Motor DQs were obtained at each outpatient NICU clinic visit over four consecutive visits. Infants who had four consecutive outpatient visits were included to trend their neurodevelopment over time (*n* = 37, 29 AGA, and 8 SGA). ADP assessments were analyzed using independent samples t-test between VLBW SGA and VLBW AGA groups for anthropometric and body composition. Repeated measures analyses of variances were completed across four outpatient visits and for weight, length, head circumference, adjusted age CAT DQ, adjusted age CLAMS DQ, adjusted age gross motor DQ between VLBW SGA and VLBW AGA groups with Bonferroni adjusted pairwise comparisons, (i.e., α = 0.05 ÷ number of comparisons).

Very low birth weight infants were started on either mother’s own milk or donor breast milk per feeding protocol on admission with parental nutrition supplement until they reach full enteral feeds (160 mL/kg/day). Enteral feeds were initiated within six hours of admission if the infant remained clinically stable. Infants were fortified to 24 kcal/oz on day of life 5–12 and reached full enteral feeds of 160 mL/kg/day by day 7–16 of life, corresponding to birth weight feeding protocol (Appendix A). Infants were discharged home on enteral feeds of mother’s own milk fortified to 24 kcal/oz with human milk fortifier or formula (Similiac^®^ Neosure, Abbott Nutrition, https://abbottnutrition.com/ (accessed on 6 June 2022)) with fortification of 24 kcal/oz or 27 kcal/oz at clinician discretion based on in hospital growth patterns. This project was designated as exempted research by the MUSC Institutional Review Board and consent was not required to be obtained.

## 3. Results

The demographics of all infants included in our study are shown in Table 1. Of the total cohort, the average birth weight was 1124 ± 183 g, and gestational age was 30 ± 2 weeks. A total of 57 infants had serial anthropometric data from birth, two ADP assessments at 44 ± 8 weeks postmenstrual age (PMA), and the second ADP assessment average was 67 ± 6 weeks PMA. Of these infants, 36 were consecutively followed outpatient for neurodevelopmental testing. Of the final 36 infants, the average gestational age was 30 ± 2 weeks and birth weight 1135 ± 185 g. At the time of discharge, 6 of 8 (75%) of SGA infants were discharged on maternal breast milk fortified to 24 kcal/oz with human milk fortifier, and 19 of 28 (68%) of the AGA cohort were discharged on the same fortified maternal breast milk regimen.

Anthropometric results from birth to ADP evaluations are shown in Table 2. There was a statistically significant difference in weight gain (grams/day) between SGA and AGA cohorts from birth to the first ADP assessment (*p* < 0.05); however, there was no statistical difference in weight gain between the two groups between the first and second ADP. Linear growth and head circumference growth were not statistically significant between the SGA and AGA groups over time as shown in Table 2.

Fifty-seven VLBW infants had serial body composition assessments which are displayed in Table 3. At the first ADP assessment, there was a statistically significant difference between percent fat (14.3% vs. 18.8%) and percent lean mass (85.7% vs. 81.2%) between the AGA and SGA groups (*p* < 0.05). However, at the second ADP assessment, there was no longer a statistical difference between the SGA and AGA body composition as the SGA cohort of VLBW infants increased fat from 14.3% to 19.2%. 

Table 4 shows the neurodevelopmental and anthropometric data across four outpatient visits of the VLBW SGA and AGA groups (*n* = 37, *n* = 8 SGA, *n* = 29 AGA). The average adjusted age from visits 1, 2, 3, and 4 were 3.0 ± 1.5 months, 7.0 ± 2.7 months, 11.7 ± 3.4 months, and 17.3 ± 3.8 months, respectively. There was a statistically significant difference between the weight of SGA and AGA cohorts at the first outpatient visit as the SGA group was smaller; however, we did not see the difference between the two cohorts at visits 2–4. There was statistical significance between the head circumferences of the two groups across all outpatient visits; however, we did not see a difference between the cohort’s linear growth. We did not see a significant difference between the adjusted CAT, adjusted CLAMS, and adjusted gross motor DQs for each of the outpatient follow-up visits between the two groups. 

## 4. Discussion

These data support previously reported findings, specifically, that a healthy cohort of SGA VLBW infants start with less body fat than AGA infants but after 50–60 weeks of PMA, the differences in fat are minimized between the two groups [16,17,18].

Premature infants accumulate body fat after birth, and at term equivalent age, they have higher fat mass and less fat-free mass than their term counterparts [18,19,20,21]. Air displacement plethysmography (ADP) is a non-invasive validated method to obtain infant body composition in the premature infant population [22,23]. Studies show that around 50–60 weeks postmenstrual age (PMA) body fat seems to stabilize in premature infants [17,18]. Our data supports this, as we did not see a significant difference between the SGA and AGA cohorts at the second ADP assessment which was around 67 weeks PMA.

SGA infants have lower body fat percent at birth and show recovery of adiposity over time compared to AGA counterparts [16,21,24]. Roggero et al. have shown that preterm SGA infants have less body fat than AGA counterparts at birth, however, after three months have similar body composition to AGA infants [16]. Our data support the findings along with these previous studies, adding information about the SGA phenotype over time.

Our study adds to the literature on VLBW neurodevelopment with respect to growth. The infants followed had repeated neurodevelopmental testing, anthropometric assessments, and serial body composition assessments. These infants were specifically selected to represent an optimized sample of infants who did not have necrotizing enterocolitis, severe retinopathy of prematurity or grade III-IV intraventricular hemorrhage and represent a sample of healthy VLBW infants without significant co-morbidities.

From birth to the first anthropometric assessment, there was a statically significant difference in weight gain (grams/day) between the SGA and AGA cohorts (*p* = 0.003). Our data also demonstrated a significant difference in weight at the first outpatient visit where the mean age was 3.0 ± 1.5 months. This represents that the SGA cohort’s catch-up growth occurs between birth to around 3 months of adjusted age. As after the second ADP assessment (mean 67 weeks PMA) and between subsequent outpatient visits 2–4, there were no significant differences in growth (grams/day) and weight between SGA and AGA groups. Although the SGA cohort had significantly smaller head circumference than the AGA group, this did not impact their neurodevelopmental testing as we saw no difference across the Capute Scales CLAMS DQ, CAT DQ, and Gross Motor DQ for adjusted age. Linear growth between the SGA and AGA cohorts was also not significant from birth to outpatient follow-up.

The Capute scales have been noted for ease and speed of administration, which make the CAT/CLAMS a reasonable choice for assessment of early development by pediatric health care providers [25,26] as well as its high test-retest reliability in preterm infants [15]. The CAT/CLAMS uses a developmental quotient (DQ) which is calculated on a ratio formula of developmental age divided by chronologic or adjusted age multiplied by 100 [26]. The test gives a mean score of 100 with standard deviation of 15. The threshold used for developmental delay is a score of <70 (2 standard deviations below the mean) [27]. The Peabody Developmental Motor Scales is another instrument for measuring motor abilities which has been shown to have high test and re-test reliability in the VLBW population [28]. Of note, none of the infants in either group were below cut off (<70) for developmental delay which may have added to our above average testing scores. In a meta-analysis performed by Sacchi et al., SGA preterm infants across multiple studies had worse neurodevelopmental outcomes compared to AGA infants [7]. We suggest that poor neurodevelopmental outcomes seen in SGA infants may be more associated with co-morbidities than SGA status itself, as we were unable to appreciate poor neurodevelopmental outcomes in this cohort. These studies demonstrate the importance of early intervention to promote neurodevelopment in this vulnerable population. 

Limitations of this study include the observational retrospective nature of this cohort. With this data review, some infants did not have serial ADP assessments or were not seen in serial outpatient visits, limiting the number of infants seen consistently in follow-up (37 of 92, 40%). Inconsistent follow-up was due to missed outpatient appointments or if infants did not need further interventional services from the clinic. We were unable to correct for maternal IQ or educational level, which was unknown in this cohort of infants. Given the structure of our NICU follow-up clinic, most infants at the time of outpatient follow-up were referred to South Carolina’s home therapy programs and most infants were involved in physical and occupational therapy services as a standard of care, which may have benefitted early neurologic testing scores. Lastly, our outpatient clinic utilizes the Capute Scales instead of Bayley for neurodevelopmental testing due to ease of administration with a shorter amount of training compared to the Bayley examinations. Additionally, our center promotes intensive post-discharge fortification of breastmilk and formula to 24 kcal/oz which is not a standard practice universally. More studies are needed on the association between discharge nutrition and neurodevelopmental outcomes. Moving forward with this research, observing a modern cohort of SGA VLBW and AGA VLBW infants utilizing the Bayley Infants Scales is warranted. Future studies should also be powered to detect a statistical difference between body fat and neurodevelopmental outcomes. Although modern nutritional changes for preterm infants have not substantially changed since this data collection, it would be prudent to study a modern cohort of infants prospectively.

It is possible there was no difference between neurodevelopment across the two groups given the SGA cohort grew as well as the AGA cohort with early catch-up growth demonstrated by significant body fat accrual and weight gain of the SGA cohort before 60 weeks PMA. Furthermore, the increase in body fat and early weight gain in this SGA population may play a role in neuroprotection as alluded to in Okada et al. [10] as we saw no differences in neurodevelopmental scores. Overall growth trajectories and early body fat accrual may positively impact VLBW neurodevelopment.

## 5. Conclusions

The trajectory for catch-up growth of VLBW infants that reflects optimal neurodevelopment is still unknown. Our data reflects trends within a healthy preterm SGA cohort, whose increasing fat “catch-up” growth was not associated with the previously reported sub-optimal outcomes seen in this preterm SGA group and was comparable to the AGA group. Further studies should be completed to investigate the associations between body composition of preterm SGA infants, specifically adiposity within the NICU, neurodevelopment, and late-onset metabolic consequences.

## Figures and Tables

**Table 1 nutrients-14-03051-t001:** Demographics of VLBW AGA and SGA infants (*n* = 92).

Demographic	Subgroup	*n* of 92
Sex	Male	43 (46.7)
	Female	49 (53.2)
Race	African American	59 (64.1)
	Caucasian/Hispanic	33 (35.9)
Growth classification	Small for gestational age	29 (31.5)
	Appropriate for gestational age	63 (68.4)
Nutrition	Days of TPN	12.9 ± 7.5
	Breast milk feeding during hospitalization	85 (92)
	Breast milk feeding at discharge	47 (51)

Data displayed as count with percentage in parenthesis or mean ± standard deviation.

**Table 2 nutrients-14-03051-t002:** SGA and AGA anthropometrics from birth to ADP evaluations (*n* = 57).

Anthropometrics	Time	SGA (*n* = 14)	AGA (*n* = 43)	*p*-Value
Weight gain (grams/day)	ADP1	19.8 ± 5.1	23.9 ± 4.1	0.003
	ADP2	19.9 ± 2.8	20.9 ± 3.0	0.3
Length (cm/week)	ADP1	1.0 ± 0.3	0.9 ± 0.2	0.5
	ADP2	0.7 ± 0.1	0.8 ± 0.1	0.6
Head circumference (cm/week)	Birth to discharge	0.8 ± 0.3	0.8 ± 0.3	0.8

Small for gestational age infants (*n* = 14), appropriate for gestational age infants (*n* = 43) of 57 infants with serial ADP assessments. Data displayed as mean ± standard deviation with corresponding *p* values. ADP1 = the time from birth to first ADP evaluation, ADP2 = the time between first to second ADP assessment. The average time of the serial ADP assessments was 44 ± 8 weeks and 67 ± 6 weeks postmenstrual age, respectively.

**Table 3 nutrients-14-03051-t003:** Body composition of SGA and AGA over time (*n* = 57).

ADP	Body Composition	SGA (*n* = 14)	AGA (*n* = 43)	*p*-Value
ADP1	% Fat	14.3 ± 5.2 [9.4–23.5]	18.8 ± 4.8 [11.7–34.1]	0.004
	% Lean	85.7 ± 5.2 [76.5–90.6]	81.2 ± 4.8 [65.9–88.3]	0.004
ADP2	% Fat	19.2 ± 5.9 [9.7–30.8]	17.8 ± 5.0 [4.0–29.0]	0.37
	% Lean	80.8 ± 5.9 [69.2–90.3]	82.2 ± 5.0 [71.0–96.0]	0.40

Data displayed at mean ± standard deviation with respective range of data displayed in brackets and corresponding *p*-values for each ADP evaluation. ADP1 = first air displacement plethysmography assessment (mean 44.0 ± 8.1 weeks postmenstrual age). ADP2 = second air displacement plethysmography assessment (67.1 ± 6.1 weeks postmenstrual age). Body composition displayed as percentage fat or lean mass.

**Table 4 nutrients-14-03051-t004:** Growth and neurodevelopmental scores of SGA and AGA over serial outpatient visits.

Variable	Visit	SGA	AGA	95% Confidence Interval	
				SGA	AGA
Weight (kg)	1 *	4.6 ± 0.3	5.7 ± 0.2	3.9–5.2	5.3–6.0
	2	6.4 ± 0.4	7.8 ± 0.2	5.5–7.3	7.3–8.2
	3	7.7 ± 0.6	9.3 ± 0.3	6.6–8.9	8.7–9.9
	4	9.9 ± 0.6	10.0 ± 0.3	8.7–11.2	10.2–11.5
Height (cm)	1	56.6 ± 1.1	59.5 ± 0.6	54.4–58.9	58.3–60.7
	2	63.9 ± 1.5	67.7 ± 0.8	60.9–66.9	66.1–69.3
	3	70.4 ± 1.8	74.4 ± 1.0	66.6–74.1	72.5–76.4
	4	78.0 ± 1.6	81.0 ± 0.8	74.8–81.2	79.4–82.7
Head circumference (cm)	1 *	37.2 ± 0.5	40.7 ± 0.2	36.2–38.2	40.2–41.2
	2 *	41.3 ± 0.5	44.1 ± 0.3	40.2–42.4	43.5–44.7
	3 *	43.6 ± 0.5	46.1 ± 0.3	42.5–44.6	45.5–46.7
	4 *	45.3 ± 0.5	47.6 ± 0.3	44.2–46.4	47.0–48.2
Adjusted CAT DQ	1	124.8 ± 9.4	113.8 ± 5.0	105.6–143.9	103.5–124.0
	2	96.8 ± 6.4	109.2 ± 3.4	83.8–109.8	102.3–116.2
	3	102.5 ± 6.1	105.5 ± 3.3	90.1–114.9	98.9–112.1
	4	97.4 ± 7.6	104.8 ± 4.1	82.0–112.8	96.5–113.0
Adjusted CLAMS DQ	1	131.4 ± 12.8	135.0 ± 6.8	105.4–157.3	121.0–148.8
	2	101.0 ± 7.8	108.4 ± 4.2	85.2–116.8	99.9–116.8
	3	105.3 ± 6.9	99.1 ± 3.7	91.3–119.2	91.7–106.6
	4	98.8 ± 8.0	98.5 ± 4.3	82.5–115.1	89.8–107.3
Adjusted gross motor DQ	1	112.6 ± 5.4	124.3 ± 10.2	101.5–123.6	103.6–144.9
	2	99.9 ± 3.7	95.0 ± 6.9	92.3–107.4	80.9–109.1
	3	103.1 ± 3.7	103.9 ± 6.8	95.7–110.5	90.0–117.8
	4	105.8 ± 3.6	106.8 ± 6.7	98.5–113.0	93.2–120.3

Adjusted developmental quotients correspond to adjusted age at time of developmental visit. Adjusted age is equivalent to chronologic age adjusted for prematurity. Data displayed as mean ± standard error with confidence intervals. Mean adjusted age in months with standard deviation of each outpatient visit 1–4 is 3.0 ± 1.5, 7.0 ± 2.7, 11.7 ± 3.4, 17.3 ± 3.8, respectively. * Statistically significant confidence intervals for each outpatient visit.

## Data Availability

Data are available by request to the primary corresponding author at lach@musc.edu.

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
