# Peer review of "Body Composition and “Catch-Up” Fat Growth in Healthy Small for Gestational Age Preterm Infants and Neurodevelopmental Outcomes"

_nutrients, 2022, doi:10.3390/nu14153051_

Round 1
Reviewer 1 Report
GENERAL COMMENT
This is a retrospective observational study that examines growth, body composition and neurodevelopmental outcomes in a cohort of “healthy” SGA and AGA VLBW infants.
Due to several methodological limits, the study results are likely to be unreliable: the report lacks essential information (according to the STROBE 2010 checklist - attached), so the paper needs to undergo major revisions.
SECTIONS
- INTRODUCTION
The introduction is sufficiently thorough and the references are updated. The topic is of great interest in the neonatological field.
- METHODS
Exclusion criteria. Bronchopulmonary dysplasia is not mentioned among the exclusion criteria: was it an exclusion criterion?
There’s no mention of how the sample size has been determined.
In the “Materials and methods” section the timing of the anthropometric measurements and neurodevelopmental tests are not reported
It is questionable to use CAT/CLAMS and Capute as a diagnostic tools: they should be used rather as a screening tests. In particular CAT/CLAMS “permits the physician to formulate an accurate clinical impression of possible mental retardation and to make referrals for definitive diagnosis”. (Kube DA, Wilson WM, Petersen MC, Palmer FB. CAT/ CLAMS: Its use in detecting early childhood cognitive impairment. Pediatr Neurol 2000;23:208-215.). The use of these tests makes difficult to compare the results with most of the studies available in the literature.
- RESULTS
Are the subjects for which neurodevelopmental data are available 36 (as in Results line 93) or 37 (as in all other sections, table 3 included)?
The major limit of this study is represented by the smallness of the final population (37 subjects, representing 64.9% of ADP population and 40.2% of the initial population).
The reasons why only 57/92 selected subjects underwent the ADP are not explained, nor the reasons for the losses at follow up (20/57, that is 35.1%). Consider use of a flow diagram to clarify.
What are the characteristics of the subjects lost to follow up? Are they similar to those of the analyzed group? Otherwise the results are not generalizable
The SGA infants analyzed are only 8: the title should therefore be modified "Body composition, growth and neurodevelopmental outcomes in a cohort of healthy SGA and AGA VLBW infants."
Comparing AGA and SGA it would be better to use the weight gain rate (g / kg / day) rather than the weight gain (g / day): it is quite predictable that the SGA, especially in the first postnatal phase, have a lower weight gain than AGA
Table 1. The number of subjects does not appear in the header (N = 92)
Tables 2 and 3: please report number of AGA and SGA in the header and not in the legend
There are 57 subjects undergoing ADP, while those with growth data and neurodevelopmental scores are only 37. It would be useful to report in a further table, the data on the body composition of the latter 37 subjects, separately by SGA and AGA
- DISCUSSION AND CONCLUSION
The discussion is thorough, but, as the authors themselves declare, the main limitation of the study is the low number of subjects seen consistently in follow up, which makes the generalizability of the results questionable, in particular on SGA infants (N = 8). This strong limitation should be further stressed in discussion.
It would have been interesting to correlate the body composition results derived from ADP with the neurodevelopmental outcome, which is impossible, since the populations are not the same. This limitation also needs to be stressed in discussion.

Reviewer 2 Report
The review article (#nutrients-1820455) by Laura Lach et al. Body composition and ‘catch-up’ fat growth in small for gestational age preterm infants and neurodevelopmental outcomes.
The paper presents a very important health problem in SGA and VLBW children, which are linked with a very high effort for physicians and emotional for parents. The current life style leads to a premature labor. The consequence of it is SGA children and catch-up, phenomenon with numerous side effects later in life. However these side effects are not fully known as it is presented in many animals studies. On the other hand, there is additional problem with VLBW infants delivered at time, but they weigh very low.
1. L 11 – the abstract cannot include abbreviations which are not explained.
2. The introduction is compressive and presents the issue in a very good manner.
3. L 52-53 – it is not clear. Both VLBW and SGA infants weigh very little at birth. If Authors want to called SGA infants VLBW They have to explain because normally SGA and VLBW are considered two separate health problems. SGA are always very small and additionally their prenatal growth is interrupted.
4. Are VLBW infants AGA?
5. L 78 – it should be mentioned that this description relates to both SGA and VLBW infants or it should be explained that the study involved only SGA as it is written in L 99
6. The infants` nomenclature should be unified. Now it is very chaotic. The reader must follow smoothly.
7. Table 4. To better understanding the table, the time when the visit was performed have to be added
